# Maker Math: Exploring Mathematics through Digitally Fabricated Tools with K–12 In-Service Teachers

**Jason R. Harron [1,*], Yi Jin [1], Amy Hillen [2], Lindsey Mason [3] and Lauren Siegel [4]**

[1]  School of Instructional Technology and Innovation, Kennesaw State University, Kennesaw, GA 30144, USA
[2]  Department of Elementary and Early Childhood Education, Kennesaw State University, Kennesaw, GA 30144, USA
[3]  Dean Rusk Middle School, Cherokee County School District, Canton, GA 30115, USA
[4]  MathHappens, 1606 Pearl Street, Austin, TX 78701, USA
[*]  Correspondence: jharron@kennesaw.edu

**Abstract:** This paper reports on nine elementary, middle, and high school in-service teachers who participated in a series of workshops aimed at exploring the wonder, joy, and beauty of mathematics through the creation and application of digitally fabricated tools (i.e., laser-cut and 3D printed). Using the Technological Pedagogical and Content Knowledge (TPACK) framework to investigate technological, pedagogical, contextual, and content knowledge, researchers applied qualitative methods to uncover the affordances and constraints of teaching and learning math concepts with digitally fabricated tools and examined how the workshops supported broadening participation in mathematics by focusing on the connections between mathematical inquiry, nature, and the arts. Affordances include opportunities for hands-on learning, visual support at the secondary level, and real-world connections that go beyond the state standards. Barriers include purchasing a laser-cutter, ventilation and noise issues, time constraints, misalignment with school and district priorities, and a lack of administrative support. All participants indicated that they were interested in additional workshops focused on designing their own digitally fabricated mathematics tools that better align with their grade level(s) and standards.

**Keywords:** aesthetics; digital fabrication; in-service teachers; maker education; professional development; TPACK

**MSC:** 97B50; 97U60

## 1. Introduction

How can teachers integrate the wonder, joy, and beauty of mathematics into their students' daily mathematical learning experiences? Starting in 2018, the National Council of Teachers of Mathematics (NCTM) began publishing their *Catalyzing Change, Initiating Critical Conversations* series, which aims to "initiate the critical conversations on policies, practices, and issues that impact mathematics education" [1] (p. ix). This three-book series offers four key recommendations that must be enacted to support positive change in K–12 mathematics education. These recommendations include: (a) broaden the purposes of learning mathematics, (b) create equitable structures in mathematics, (c) implement equitable mathematics instruction, and (d) develop deep mathematical understanding [1–3]. Of particular interest to this study is the first recommendation, which emphasizes that students should "understand and critique the world [through mathematics], and experience the wonder, joy, and beauty of mathematics" [4]. While this recommendation is a laudable goal, NCTM also points out that "for children to experience the wonder and joy in learning mathematics so must their teachers" [3] (p. 21).

Mathematics, nature, and beauty have historically been intertwined, with mathematical discourse frequently emphasizing the aesthetic pleasure that can be derived from the

elegance, harmony, and order found in numbers, patterns, and shapes [5,6]. The beauty of mathematics takes many forms, including the divine beauty of the golden ratio and Fibonacci sequence found in the arrangement of seeds in a sunflower [1,7]; the natural beauty of the Voronoi structures located in leaf cells, giraffe markings, and dragonfly wings [8]; and the elegant beauty that exists in the simplest and most charming mathematical proofs [9]. Yet, despite the natural connection between aesthetics and mathematics, K–12 (K–12 is a short form for the publicly supported school grades prior to college. These grades are kindergarten (K) and the 1st through the 12th grade (1–12)) education fails to acknowledge the relationship between mathematical inquiry, nature, and the arts [5] and lacks the meaningful, real-world connections required to foster a lifelong appreciation of the wonder, joy, and beauty of mathematics [10,11]. Therefore, researchers and teacher educators need to support practitioners by expanding their conceptualization of mathematics [12].

One possible pathway towards this goal is through the emerging field of Science, Technology, Engineering, Arts, and Mathematics (STEAM) education. According to Mejias and his colleagues [13], STEAM "represents ongoing attempts by educational researchers, practitioners, and policymakers to make sense and potentially institutionalize the role of the arts in relation to science, technology, engineering, and mathematics [STEM] learning" (p. 210). Thriving at the intersection of new media, making, and digital technologies [13–15], STEAM education has the potential to help teachers and students develop a deeper appreciation for mathematics in a way that is active, integrated, and connected to the world around them. Despite the growing interest in STEAM education, there are limited empirical data on how to prepare teachers to effectively engage in STEAM-based instructional strategies [16,17]. While the outcome of STEAM teacher professional development (PD) has been shown to increase students' creativity and motivation [18,19], teachers wanting to implement STEAM-based instruction continue to face numerous challenges related to lesson planning, pacing/time, student comprehension of content, district policies, technology integration, and standardized assessment [17].

While the transition from STEM to STEAM continues to be hotly debated [13,18], there is evidence that digital fabrication technologies (e.g., 3D printers and laser-cutters) can serve as a catalyst for the successful integration of STEAM education [20,21]. Building on the influence of the technologies adopted by the Fabrication Laboratory (FabLab) at MIT [22], tools such as 3D printers and laser-cutters are beginning to find their way into K–12 schools [21,23,24] and teacher preparations programs around the globe [25,26]. Simply providing teachers with access to technology, however, is not enough to guarantee adoption. Rather, prior research suggests that teachers' pedagogical beliefs and self-efficacy play an important role in whether technology is integrated into the classroom [27–29]. While digital fabrication technologies are thriving in engineering classrooms and makerspaces, there is limited research on how they can be used to create hands-on tools, such as mathematical manipulatives, to support teaching and learning. Therefore, the purpose of this study was to examine how digitally fabricated mathematics tools can be used to help in-service teachers explore the wonder, joy, and beauty of mathematics through the authentic connections that exist between mathematics, nature, and the arts.

## 2. Literature Review

To better understand how teachers perceive digitally fabricated tools in the context of mathematics, the following section reviews the literature associated with making, digital fabrication, and the role of technology in K–12 mathematics education. This section also includes an overview of the Technological Pedagogical and Content Knowledge (TPACK) framework and reviews research about the impact of PD on mathematics teachers' TPACK.

### 2.1. Making and Digital Fabrication in K–12 Education

Over the past decade, there has been increasing interest in how K–12 education can benefit from the tools, practices, and mindsets of the maker movement. The maker movement, as defined by Halverson and Sheridan, "refers broadly to the growing number of

people who are engaged in the creative production of artifacts in their daily lives and who find physical and digital forums to share their processes and products with others" [30] (p. 496). While thriving in informal educational settings (e.g., afterschool programs, summer camps, and community makerspaces), making and the maker movement serve as an outlet for do-it-yourself (DIY) hobbyists and tinkerers to connect with other creative individuals through the sharing of knowledge and showcasing of their creations [31]. Grounded in the learning theory of constructionism [32], which states that learning is best made visible through the creation of physical and digital artifacts that can be reflected upon and shared with others [33], maker education is a student-centered approach that can be used to empower individuals through the examination and design of everyday objects, ideas, and systems [34].

The technologies and tools associated with the maker movement are commonly divided into no-tech, low-tech, and high-tech categories. No-tech tools may include materials that are often associated with arts and crafts (e.g., knitting needles, cardboard, crafts sticks, and copper tape), while low-tech tools typically run on electricity and can be semi-automated (e.g., woodworking tools, soldering irons, and sewing machines). High-tech tools for making are computational in nature and include single-board microcontrollers (e.g., Arduino, micro:bit, and Raspberry Pi) and digital fabrication tools (e.g., 3D printers, computer numerical control (CNC) machines, and laser-cutters). While many of these technologies are still emerging and remain nascent in K–12 education, they have begun to find their way into school makerspaces, FabLabs, media centers, and shop classes [35].

The most common digital fabrication technology that has been adopted in K–12 education is 3D printing. In recent years, consumer-grade 3D printers have become relatively affordable, safe, and reliable, making it possible for schools to purchase one or more printers for students and teachers to use for educational purposes. Examples of using 3D printing in K–12 education include introducing middle school girls to digital fabrication through computer-aided design (CAD) [36], creating replicas of fossils for hands-on paleontology experiences [37], and helping students explore their career interests in biomedical engineering [38]. More recently, consumer-grade laser cutters have also been introduced to K–12 education. Unlike 3D printing, which uses an additive manufacturing process by extruding melted plastic filament, laser cutters use a subtractive process by focusing a beam of light to cut and etch materials such as plywood, cardboard, and acrylic plastic. While laser cutting is much faster than 3D printing, it requires more safety precautions due to the risks associated with Class 4 lasers and the fumes produced by the cutting process [39]. While laser cutters are often used in engineering classes to create gears and boxes [40,41], they can also be used to produce hands-on manipulatives such as acrylic tiles to introduce block-based coding [42], 3D model kits to explore molecular geometry [43], and foldable origami forceps to learn about pincers and tweezers used in microscopic surgery [44].

Furthermore, internationally, there has been momentum for introducing digital fabrication, particularly FabLabs, at the primary and secondary levels to explore how mathematics can be more broadly and harmoniously contextualized across the spectrum of nature and the arts [45]. As such, more opportunities are emerging for teachers and students to learn how to use these technologies as part of the educational process.

*2.2. The Role of Technology in Teaching and Learning Mathematics*

As noted in NCTM's *Principles to Actions: Ensuring Mathematical Success for All*, "tools and technology must be indispensable features of the classroom" [12] (p. 78). Appropriate use of tools and technology supports students' learning, reasoning, and communication about mathematical ideas.

In the elementary classroom, mathematical tools often come in the form of hands-on manipulatives [12]. Mathematical manipulatives are defined as "any tangible object, tool, model, or mechanism that may be used to demonstrate a depth of understanding, while problem-solving, about a specified mathematical topic or topics" [46] (p. 184). These physical objects serve as an instructional tool allowing students to explore concepts, demonstrate

understanding, and develop numerical skills. Unlike static images or diagrams, manipulatives allow users to flip pieces, group objects, and build figures. Manipulatives used in mathematics classrooms take many forms, including counters, snap cubes, base-ten blocks, tangrams, geoboards, and algebra tiles [12,47].

As students mature, technology in the mathematics classroom typically transitions from hands-on manipulatives towards the use of 10-key, scientific, and graphing calculators. For example, TI-84 graphing calculators have been commonly used to support function modeling in the classroom [48] and enable the generative exploration of mathematics through participatory simulations [49,50]. In recent years, advances in digital technologies have led to the development of online and application-based tools that serve as virtual manipulatives on smartphones, laptops, and tablet computers. Virtual manipulatives are "an interactive, web-based visual representation of a dynamic object that presents opportunities for constructing mathematical knowledge" [51] (p. 373). These can include online graphing calculators and web applications such as Desmos and GeoGebra. Using such "mathematical action technologies" [52], students are able to interact with mathematical ideas while making and testing their own conjectures.

The role of technology in mathematics teaching and learning has also been viewed as paramount during the pandemic. According to a recent international survey on future themes in mathematics from before and during the pandemic, "Students do not only need to learn to use technology; the technology can also be used to learn mathematics (e.g., visualization, embodied design, statistical thinking). New technologies such as 3D printing, photo math, and augmented and virtual reality offer new opportunities for learning" [53] (p. 9). With the increased availability of 3D printers and laser cutters, there is the potential for a revival of teachers and students creating their own mathematical manipulatives. For this transition to take place, however, teachers need examples of how these technologies can be integrated into their teaching practices while also addressing the intersecting domains of their technological, pedagogical, contextual, and content knowledge.

*2.3. The TPACK Framework*

The Technological Pedagogical and Content Knowledge (TPACK) framework is a widely adopted framework for conceptualizing and examining teachers' needed knowledge domains when they are teaching with technology (see Figure 1, [54,55]). This framework originated from the Pedagogical Content Knowledge (PCK) framework, which illustrates the interplay of content knowledge and pedagogical knowledge during teaching [56,57]. By adding technology knowledge, TPACK scholars reconceptualized the PCK framework in consideration of the modern contexts of teaching and learning [58]. The TPACK framework is important because it helps researchers and educators conceptualize what knowledge teachers need when they integrate technology into their classrooms [59]. It also facilitates conversations among scholars and practitioners by providing them with a common language. Furthermore, researchers have been using the TPACK framework to undergird their studies for over 15 years, resulting in significant empirical findings for the field of teacher education and PD [60,61].

As shown in Figure 1, the current TPACK framework has four foundational knowledge bases: content knowledge (CK), pedagogical knowledge (PK), technological knowledge (TK), and contextual knowledge (XK) and four overlapping knowledge domains: pedagogical content knowledge (PCK), technological content knowledge (TCK), technological pedagogical knowledge (TPK), and technological pedagogical content knowledge (TPACK). Below is a list of the definitions of these constructs [58,62,63]:

- CK: Knowledge about the subject matter;
- PK: Knowledge about the methods and processes of teaching;
- TK: Knowledge about various technologies that can be applied to education;
- PCK: Knowledge of the pedagogical approaches appropriate for teaching a given content;
- TCK: Knowledge of how technology can create new representations for specific content;

- TPK: Knowledge of how various technologies can be used in teaching and understanding that using technology may change the way teachers teach;
- TPACK: Knowledge required by teachers for integrating technology into their teaching in any content area;
- XK: Knowledge required by teachers about the local and far-reaching affordances and constraints of teaching with technology.

This study uses TPACK as a conceptual framework to guide thematic coding.

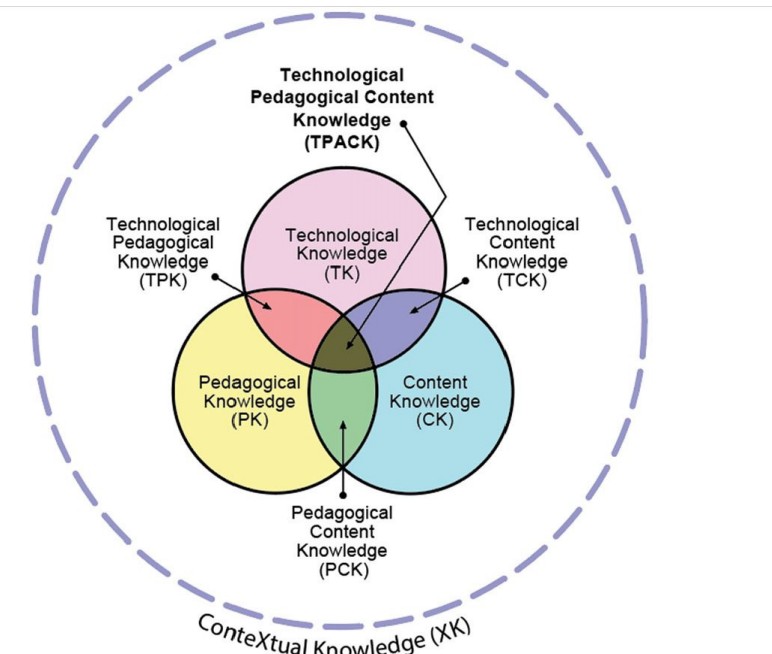

**Figure 1.** The New and Improved TPACK model. Revised version of the TPACK image. © Punya Mishra, 2018 [59]. Reproduced with permission.

### 2.4. The Impact of PD on Mathematics Teachers' TPACK

Technology integration has become an indispensable part of math education [64]. More recently, NCTM provided the following statement to highlight the importance of technology integration into math instructions:

> "Strategic use of technology in the teaching and learning of mathematics is the use of digital and physical tools by students and teachers in thoughtfully designed ways and at carefully determined times so that the capabilities of the technology enhance how students and educators learn, experience, communicate, and do mathematics. Technology must be used in this way in all classrooms to support all students' learning of mathematical concepts and procedures, including those that students eventually employ without the aid of technology. Strategic uses support effective teaching practices and are consistent with research in teaching and learning." [65] (p. 1)

To effectively use technology in the classrooms, math teachers need to be prepared and continue to develop their TPACK throughout their careers. Researchers have been using the TPACK framework as a foundation to investigate math teachers' knowledge domains and found that the lack of TPACK was a key barrier to math teachers' technology integration for student learning [66–68]. Further research suggests that more PD should focus on conceptual understanding, teaching strategies, time, and student engagement that support teachers' current pedagogical methods and then build on the list of new ideas for pedagogy [69–71].

Scholars also reported that after participating in the technological PD, math teachers still did not always use available technology during their classroom instruction [66,72,73].

To further investigate this issue, Young analyzed the results from 13 meta-analyses and identified five key considerations for improved math teaching with technology, which are duration, assessment, instructional modality, grade level, and mathematics subject matter [74]. He suggested that these factors aligned with the TPACK framework. He recommended the researchers use the TPACK framework for PD research and cautioned educators to assess the constructs that are most salient in the given context.

Young and his colleagues continued to work on this line of research. In another study, they investigated whether PD impacted math teachers' perceptions of their TPACK [75]. They found that PD in urban schools increased math teachers' PK, TK, PCK, and TCK. They believed that insufficient resources and PD in urban schools contributed to the lack of teachers' TK, which became a new form of the digital divide. Therefore, they suggested that technology PD should be encouraged early and often. They also stated that teacher educators needed to consider the influences of teachers' learning environments before designing and implementing technology PD accordingly. Rakes and his colleagues found equivalent conclusions with 17 secondary math teacher candidates [76]. They suggest that "the 'T' in TPACK is an important and unique type of knowledge that is not automatically developed along with PCK . . . Putting the 'T' back into PCK will require explicit emphasis in the PD experience with the technology purposefully integrated with the pedagogy" [76] (p. 13). They called on teacher educators to design teacher education and PD with a direct focus on using technology for math conceptual understanding.

Additionally, Bray and Tangney asserted that a mismatch existed between the researched-based best practices for math technology integration and teachers' actual implementation in the math instructions [77]. Polly conducted a year-long PD program to help elementary teachers integrate technology-rich tasks into their math instructions. Although the participants had some degree of technology integration in their classrooms and developed their TPACK, Polly found the misalignment between their enacted pedagogies and those emphasized during the PD sessions [78]. He also discovered that co-planned lessons had richer technology integration and higher TPACK enactment than those independently planned by the teachers. Polly and Orrill reported similar outcomes in another study that grades 4–8 math teachers increased their TK, but very few articulated clearly how to use technology in math instructions after participating in the PD program [79].

It is apparent from the literature that in-service teachers need more technology PD focusing on math instruction that is student-focused, teacher-owned, collaborative, content and theory-laden, reflective, and comprehensive [78]. Therefore, using a case study design, the current study examines the affordances and constraints of these tools and technologies in the context of K–12 mathematics education. In particular, this paper will address three research questions:

1.  In what ways do digital fabrication tools support teachers' perceptions of mathematics teaching, learning, and curriculum?
2.  What challenges do in-service teachers face when applying digital fabrication tools to mathematics in terms of teaching, learning, and curriculum?
3.  In what ways do the workshops influence teachers' conceptualization and practices of broadening participation in learning mathematics?

## 3. Methods

This case study uses a holistic design [80], with the case bounded to a series of three mathematics workshops over six weeks. Using qualitative methods, the unit of analysis includes a group of nine K–12 mathematics teachers, their social interactions, and artifacts generated during and between workshops.

### 3.1. Research Context and Workshop Design

*Maker Math: Using Laser-Cut Tools to Explore the Hidden Beauty of Mathematics* is a three-workshop series offered by three faculty members at a Southeastern R2 university with high research activity. This workshop series was designed to introduce K–12 mathematics

teachers to custom-made mathematics manipulatives (i.e., math tools) and provide opportunities for them to consider how to incorporate these tools into their teaching practice. It also helps teachers explore mathematical concepts, such as Voronoi patterns and the golden ratio, while sharing digitally fabricated tools that support the exploration of tessellating patterns, string art, origami, and conic sections. The workshops were offered on three Saturday mornings in a maker lab of the college of education during spring 2022.

All three faculty members participated in the design and implementation of the workshops. Before offering the workshops, these three faculty members met regularly to select the content and design the activities for the workshops. They first browsed the MathHappens website to choose interesting content and activities that were aligned to the K–12 math standards. Then, they communicated with the fifth author, who shipped selected maker math tools, books, and resources to the first author. Afterward, the three faculty outlined the activities and sequence of each workshop (see Appendix A), created additional math tools, and printed needed materials. They discussed and decided on the pedagogical strategies used in the workshops, such as direct instructions, hands-on projects, discussion, and instructor-facilitated reflection. In general, each activity followed a pedagogical pattern (introduction, instructor-led overview, hands-on projects, small group discussion, whole group discussion, and then instructor-facilitated reflection), and this pattern was repeated several times for the activities in a workshop. They also discussed and implemented the recruitment strategies and assigned roles to the instructors. The first author was the lead instructor. The second author was the participatory observer who took three sets of typed field notes. The third and fourth authors were the facilitators.

During the workshops, a variety of topics and activities were presented to in-service teachers. Workshop 1 had four activities: decahedron tiles, laser-cut rectangular puzzle, Voronoi flipbook, and Voronoi stained glass. Five different activities were offered in workshop 2: golden ratio transparent plexiglass, golden ratio calipers, origami chompers, icosahedron, and string art. Another four activities were provided at workshop 3: conic sections, nautilus gears, Mercator projection, and objects of constant width. Appendix A illustrates the details of the workshop activities. For each workshop, multiple materials and digitally fabricated tools were used. These materials and tools were prepared beforehand by the researchers and then given out to the participants as compensation at the conclusion of the workshops.

### 3.2. Participants

Participants were recruited through convenience sampling of in-service teachers who were enrolled in an instructional technology graduate program. Nine in-service teachers completed the workshops and voluntarily participated in this study. Among them, three are men, and six are women. One teacher was 21–30 years old, four were 31–40 years old, three were 41–50 years old, and one was 51–60 years old. Two teachers are African Americans, two are Asians, one is Hispanic or Latino/a, three are White, and one preferred not to tell. Participants' teaching experience ranged from 3 to 14 years, with an average of 7.9 years of teaching experience. Participants taught various grade levels and subjects, as shown in Table 1. Among these teachers, two made their own math manipulatives before, while seven did not. One teacher had used a laser cutter before, while eight had never used one. None of them had any experience in using laser-cut math tools, and they never used strategies that combine math and laser-cut math tools in their teaching experiences.

**Table 1.** Participants' demographic information.

| Teachers | Gender | Age Range | Ethnicity | Years of Teaching | Grade Level | Content Area(s) | Certification(s) |
|----------|--------|-----------|-----------|-------------------|-------------|-----------------|------------------|
| Scarlett | Female | 31–40 | Other | 7 | K–2 | All | Early Childhood Education |
| Mia | Female | 51–60 | White | 10 | 2nd | All | PK–5th |

**Table 1.** *Cont.*

| Teachers | Gender | Age Range | Ethnicity | Years of Teaching | Grade Level | Content Area(s) | Certification(s) |
|---|---|---|---|---|---|---|---|
| Ava | Female | 31–40 | White | 8 | 3rd | All | PK–5th |
| Riley | Female | 31–40 | African American | 8 | 6th–8th | Science | Math and Science (4–8) |
| Lucas | Male | 41–50 | African American | 8 | 6th–8th | Special Education, STEM | Special Education, Social Sciences |
| James | Male | 41–50 | White | 5 | 7th | Math | Math and Social Studies |
| William | Male | 21–30 | Asian | 3 | 8th | Math, Science | Math (6–12), Science (6–12), Engineering and Technology, ESOL, Gifted |
| Chloe | Female | 41–50 | Asian | 14 | 9th | Math (Geometry) | Secondary Math |
| Olivia | Female | 31–40 | Hispanic/Latina | 9 | 11th | Math | K–5 Elementary, 6–12 Math |

*3.3. Data Collection*

The researchers used multiple data collection methods to gather qualitative data, including observation notes, a semi-structured group interview, artifacts (images and objects), and a follow-up survey. First, during each workshop, a researcher observed the session and took observation notes using a laptop focusing on the conversations of content, pedagogy, and technology guided by the TPACK framework. In particular, the observer gathered authentic quotes and double checked with other workshop instructors for accuracy. Each packet of observation notes was six pages. Altogether, the researcher recorded 18 pages of observation notes. Second, at the end of the third workshop, the researchers conducted a group interview with all nine in-service teachers using five semi-structured interview questions (see Appendix B). The researchers took detailed notes and recorded the answers to these questions on a shared Google Doc. Afterward, they read the document and checked for accuracy. The researchers collected the images of participants' artifacts, such as their maker objects and math tools as well as the images they shot and sent to the lead researcher. Last, the researchers sent out an online follow-up survey 14 days after the workshop with a checklist and open-ended questions asking how teachers used or shared the tools, along with a table of workshop activities and pictures (see Appendix C).

*3.4. Data Analysis*

The qualitative data sources (i.e., field notes, open-ended survey questions, and participant-generated artifacts) were analyzed using a hybrid approach [81] that included both deductive coding based on the TPACK framework and inductive open-thematic coding [82]. First, the lead researcher used a collaborative document to independently code the qualitative data for the barriers and supports of integrating digitally fabricated math tools in their practice (RQ1 and RQ2). In order to facilitate data analysis, the data were split into episodes that included pre-workshop and one episode for each of the 13 digitally fabricated math tools and the final whole-group discussion. Within each episode, multiple data sources (i.e., observation notes, semi-structured group interview transcripts, and artifacts) were included to support triangulation of data. For example, multiple sources suggested that participants were experiencing limited buy-in from school culture. This included comments recorded in field notes, specific mentions regarding aligning lessons with school mission and vision, and lack of support from administrators. Table 2 shows the data sources used to answer the research questions and the methods used for data

triangulation. Following the hybrid coding approach, all data were coded using the TPACK framework with the initial codes of technological knowledge (TK), pedagogical knowledge (PK), content knowledge (CK), and contextual knowledge (XK). The data were then analyzed a second time using open-thematic coding to identify any additional themes. A second researcher then checked these coded responses, and both researchers worked together to verify, modify, and refine the codes until interrater reliability of 100% was achieved. A total of 44 codes were organized and categorized into emerging themes using a constant comparative method [83]. For RQ1, themes included supporting tactile and visual learning, real-world connection, economics, and community. Additionally, themes for RQ2 included both physical (i.e., economic, environmental, technical, time, and people) and psychological (i.e., pedagogical knowledge, student prior knowledge, and culture) barriers.

**Table 2.** Data sources and triangulation.

| Research Questions | Data Sources Used | Data Triangulation |
|---|---|---|
| 1. In what ways do digital fabrication tools support teachers' perceptions of mathematics teaching, learning, and curriculum? | • Observation notes<br>• Semi-structured group interview transcripts<br>• Artifacts and documentation | • Key findings were supported by field notes, photos, and follow-up survey. Participants reported using math tools to support hands-on learning and multiple participants reported creating more manipulatives at school. |
| 2. What challenges do in-service teachers face when applying digital fabrication tools to mathematics in terms of teaching, learning, and curriculum? | • Observation notes<br>• Semi-structured group interview transcripts<br>• Artifacts and documentation | • Challenges were identified through the triangulation of field notes, semi-structure group interviews, and follow-up survey. Findings indicate physical and psychological barriers: particularly from buy-in from existing school culture. |
| 3. In what ways do the workshops influence teachers' conceptualization and practices of broadening participation in learning mathematics? | • Observation notes<br>• Semi-structured group interview transcripts<br>• Artifacts and documentations<br>• Follow-up survey data | • Participants provided multiple forms of evidence about broadening participation in mathematics through the photos sent between workshops and discussions about sharing math tools with other teachers. |

Second, all data were transferred to a collaborative Google Sheet, so themes and episodes could be viewed side by side. Columns were setup based on the TPACK framework using the same TK, PK, CK, and XK deductive themes. Taking a hybrid approach, inductive coding was then used to identify additional themes related to the concept of broadening participation in mathematics. These new themes included aesthetics, participant examples, and misconceptions (see Appendix D). Furthermore, the spreadsheet included an additional notes column to help researchers synthesize the data. Each data source was coded by at least two researchers who constantly compared data, codes, and themes. Data were reanalyzed to resolve any disagreements until 100% interrater reliability was reached.

## 4. Results

This section reports the results of this study. It has been organized according to the research questions. In each sub-section, detailed results are presented.

### 4.1. RQ1: In What Ways Do Digital Fabrication Tools Support Teachers' Perceptions of Mathematics Teaching, Learning, and Curriculum?

Analysis of qualitative data indicated that digital fabrication tools supported teachers' exploration of mathematics teaching, learning, and curricula in various ways, such as tactile

and visual, real-world connections, economic, and community support. First, participants said that digital fabrication tools are tactile, which supports hands-on math activities focusing more on problem-solving and creativity. These tools help teachers and students better visualize mathematics concepts while developing conceptual understanding. In terms of pedagogy, all participants commented that math tools such as manipulatives were used more in elementary school but less in secondary schools. Secondary education teachers particularly highlighted the importance of play and fun in mathematics learning in the middle and high school context. They advocated for more tactile and visual hands-on activities using math manipulatives for middle and high school students.

Second, all teachers agreed that digital fabrication math tools can help students develop conceptual understanding, make real-world connections, and develop agency. Several teachers expressed the importance of teaching real-world connections to students as a way of developing a contextual understand of mathematics. For example, Chloe mentioned,

> "You can use slope and quadratics in the real world, but simplifying a rational expression is hard to explain to kids since you don't know where you will need it in the real world. But things like statistics you are going to use in the real world. Sometimes it's applicable and sometimes it's not worth talking about. Just here is the standard and let's move on."

Teachers also gave a few examples of how to connect the mathematics concepts they teach to nature and real life. Some suggested that students can find the Fibonacci sequence in flowers, and teachers can take students outside to explore. Others proposed finding Voronoi patterns in nature and architecture. One teacher proposed that they could let students search for stained glass windows in their community and then create their own designs. Scarlett stated that she will have her students use the golden ratio calipers to measure their bodies and find patterns in nature that follow the golden ratio rule. All teachers concurred with the potential of these tools in cultivating students' engagement and ownership since they will be working on the maker math activities/projects from start to finish, which involve exploration, learning mathematics concepts, and designing their own artifacts. In such activities, students will actively engage in artistic design and developing their technological skills while bringing what they create on paper to life. Participants emphasized that these design-related math activities/projects help students become makers instead of consumers.

Third, the digitally fabricated tools introduced and used in the workshops were created using inexpensive materials, such as cardboard, plywood, and filament. These materials are relatively inexpensive and easy to work with. Making math tools using these materials is more economic than purchasing commercial sets because no shipping and other fees are involved. Once teachers develop their technological knowledge needed to operate these digital fabrication tools, they can begin designing their own creations and mass-produce classroom sets for their students at a very low cost. Moreover, they will be able to teach their students how to make their tools and customize their designs.

Lastly, although the materials are inexpensive and easy to use, teachers still need school/community support to obtain digital fabrication machines, such as crafting and cutting machines, 3D printers, and laser cutters. Thus, participants reported that developing school/community support is crucial to ensuring these tools can be successfully implemented within their teaching contexts. James mentioned that an outside company helped purchase a 3D printer for his school. With eight 3D printers in total after various support efforts, he was able to apply his technological knowledge to design more 3D printing math activities for his students. Other teachers agreed on the affordances of 3D printing math activities for real-life applications. For example, students can create unique designs and make custom fidget spinners to sell in the school stores, which helps them develop both their math skills and entrepreneurship. All teachers suggested that school/community support is an indispensable part of promoting digital fabrication in K–12 schools, and ideally, every classroom should have some digital fabrication machines and materials such

as a 3D printer and filament to support the ongoing development of how these technologies can impact learning in the K–12 mathematics setting.

*4.2. RQ2: What Challenges Do In-Service Teachers Face When Applying Digital Fabrication Tools to Mathematics in Terms of Teaching, Learning, and Curriculum?*

Analysis of qualitative data identified several constraints that may impact in-service teachers' use of digital fabrication tools in their teaching practice. During the focus group interview, teachers spoke of various physical and psychological barriers to using digital fabrication tools in their practice.

The physical barriers that teachers identified included economic, environmental, technical, time, and people. Teachers shared concerns about the heavy upfront cost, amount of training needed, the requirement to purchase materials, and the space that was needed to start the integration of digital fabrication in their schools. They also discussed aspects related to the design of the learning environment, such as the location and ventilation of laser cutters, the number of math tools needed for the whole classroom, and the materials and resources each teacher needs to teach mathematics concepts in depth. Compared to the relatively low barriers of 3D printing, participants believed that laser cutting had relatively higher technical barriers for teachers and students. Time was a considerable barrier to all participants (e.g., limited instruction time for the content beyond the standards that must be covered, limited planning time, and time needed to attend meetings). Meanwhile, the pacing of their math curricula posed some challenges, as teachers had to finish teaching their curricula before the state standardized testing. Teachers also spoke of constraints on their agency. For example, Olivia noted that in her district, teachers are held accountable for using "common activity, common lessons, and common testing". Because of the heavy workload and strong emphasis on accountability, participants felt like they had no time to learn, play with, or make the math tools as well as participate in PD. As a result, the teachers expressed that it was difficult for them to develop the pedagogical and technological knowledge needed to implement digital fabrication tools in a K–12 math context. Furthermore, the preparation time of designing and offering these maker math activities gave teachers the perception of adding extra time and work to their existing workload, which made them feel less confident. The last physical barrier these teachers mentioned were people. They commented that one teacher might not be enough for the maker math activities, and it would be easier to have multiple teachers in the classrooms to help students break down the mathematical concepts in small groups.

The psychological barriers that teachers identified included pedagogical knowledge, student prior knowledge, and culture. During the focus group interview, teachers expressed that they felt they lacked enough pedagogical knowledge of how to use digitally fabricated tools in mathematics instruction. They suggested that their lack of confidence in pedagogy might prevent them from trying out new maker math activities in their classrooms. Another concern was the consistency across multiple teachers, especially the buy-in from other teachers in the same professional learning communities who taught the same curriculum. Moreover, buy-in from multiple levels in the school, such as administration, was needed. A few teachers mentioned that their school districts required common activities and lessons. This requirement also made it challenging to use creative pedagogical strategies. Besides the pedagogical knowledge, teachers believed that students did not have enough prior knowledge in connecting mathematics concepts to the real world, and they did not have enough technical skills to design and make digitally fabricated artifacts. More importantly, participants strongly advised that school culture could be a barrier to integrating digital fabrication math activities in their classrooms. For example, William said that the maker math activities might not align with his school's and district's mission and vision. A few teachers agreed on the misalignment of the maker math activities and the mission and vision of their schools and districts. They mentioned a few possible reasons for this misalignment, such as the novelty of the activities, request to purchase equipment and materials, lack of experience, no previous exposure, and conflict between the requirement

of the state and the need of the county. Again, teachers believed buy-in from people at multiple levels of the schools and districts was fundamental to promoting the digitally fabricated math activities. Therefore, teachers interested in implementing these activities need to develop a strong contextual understanding of their school and district.

*4.3. RQ3: In What Ways Do the Workshops Influence Teachers' Conceptualization and Practices of Broadening Participation in Learning Mathematics?*

In line with the goals of NCTM, the workshops aimed to broaden participants' conceptualization of the purposes of learning mathematics. This section details findings of how participants visualized mathematics at the intersection of nature and art and how the tools from the workshops could be used to broaden participation in their classrooms. Additionally, this section documents how teachers shared their tools with students, colleagues, family, and friends and provides a list of participant-generated recommendations for future workshops.

4.3.1. Visualizing Mathematics at the Intersection of Nature and Art

In total, five participants sent a total of 11 photos either via text message or e-mail between workshops. Photos fell into three categories: (a) nature, (b) visual arts, or (c) string art. Nature pictures included a mint leaf with a Voronoi pattern throughout its veins, two photos of pine cones (top and side view) depicting a three-dimensional golden ratio spiral, and four photos of orchids with the Fibonacci sequence found in the flower of the plant (one lip, three inner petals, and five outside petals). One visual art photo depicted a school mural with a tractor breaking through a wall with a Voronoi-like pattern (but the pattern was not mathematically accurate). Three photos were string art created by the participants. Of the string art, two depicted somewhat random patterns with the string radiating from a focal point or creating a zig-zag pattern. One string art included two parabolas, which formed a circular pattern (see Figure 2).

The visual aspects of the workshops were often expressed through the participants' aesthetic choices involving color and symmetry when working with the laser-cut math tools. Starting with the decahedron activity (workshop 1, activity 1), the laser-cut plywood pieces were pre-colored with water-soluble bingo daubers, so one side was either red, yellow, green, purple, or blue, and the backside was colored black. The use of color added an additional layer of symmetry to the "round" shapes, with the majority of participants focusing on the rainbow colors, and only one participant incorporating the black tiles into their creation. Similarly, participants were given a choice of colorful loom rubber bands to create the edges of their icosahedron (workshop 2, activity 4). The use of more than one color of rubber band provided generative space for participants to arrange the colors in ways that produced different types of symmetry within three-dimensional space. This desire for symmetry sparked conversations about how many rubber bands were needed to complete the shape and raised questions about the optimal number of colors needed to create the most symmetrical patterns.

Asymmetry also played an important role in the discussions about the wonder, joy, and beauty of mathematics while also serving to facilitate discussion about mathematical concepts. For example, during the Voronoi stained glass activity (workshop 1, activity 4), color played a major role in highlighting the asymmetry of their Voronoi-esque creations, with several participants spacing out their like-colored cells to avoid any two adjacent cells from having the same color. These spacing of these colors resulted in mathematical conversations between participants about minimum number of colors they would need to avoid having any like-colored cells touch. Additionally, activities involving the golden ratio spiral (workshop 2, activity 1) and calipers (workshop 2, activity 2) helped participants uncover how the visual arts and nature contain asymmetrical proportions and ratios. While golden ratios are often present in contexts such as paintings from the Renaissance, participants were more likely to see connections to the golden ratio through the use of the

calipers by measuring the portions in their body (e.g., elbow, wrist, fingertip), puppets and stuffed animals, and pictures of insects, birds, and mammals from oversized library books.

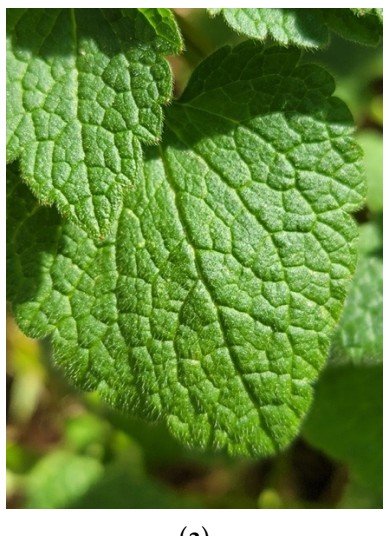
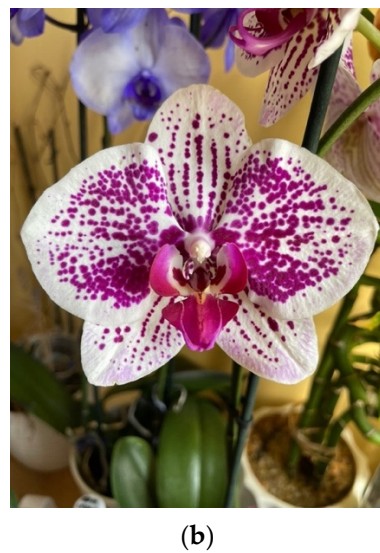
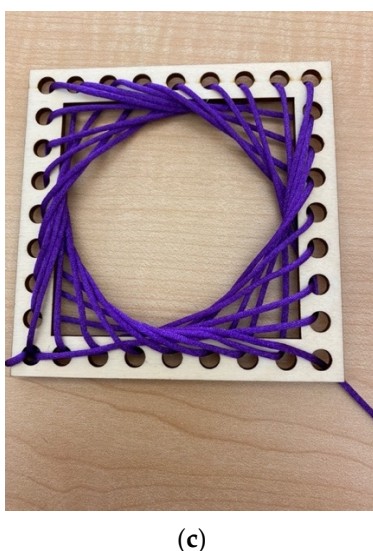

(**a**)        (**b**)        (**c**)

**Figure 2.** Participants' photos: (**a**) James—Mint leaf, Voronoi pattern; (**b**) Chloe—Orchid petals, Fibonacci sequence (one lip, three inner petals, and five outside petals); (**c**) Mia—String art, circular pattern with two parabolas (conic section or two quadratic functions).

### 4.3.2. Broadening Participation in Mathematics

While broadening participation in mathematics was one of the overarching goals of the workshops, data analysis revealed a tension between whether these activities are for all students or if they best serve specialized populations (e.g., gifted and talented students or special education). During the focus group interview, numerous participants indicated that these activities could serve as an extension or as a reward/challenge for more advanced students who finish their work early. Olivia discussed how she thought the wooden tile activities (workshop 1, activities 1 and 2) could be fun for students to use during their free time or as part of a "brain break". As part of the follow-up survey, Mia wrote, "My plan is to be a gifted teacher in a couple of years which means I'll be teaching 1st–5th graders and have more opportunities to use the math tools". This finding indicates that teachers need further development of their pedagogical knowledge to feel comfortable implementing these digitally fabricated math tools with general education students.

In contrast, several participants indicated that these activities could be used as hands-on tools to introduce new content. During the focus group interview, William mentioned that these tools could be "used as an activating strategy to introduce a topic, or as a student-led conclusion or summary". Chloe explained that she thought the conic section tool (workshop 3, activity 1) could be "used as a model to have students explain what they learned, or as an activating strategy". Additionally, as part of the follow-up survey, Chloe noted that "I plan to use the Fibonacci plexiglass and golden ratio calipers as part of my lesson during the 'Arithmetic and Geometric sequence standard' and Fibonacci numbers as an extension in Unit 9". Riley saw these tools extending the practices of the traditional mathematics classroom by stating that "in order to properly teach [mathematics], it needs to be a combination of both pen and paper and hands-on". Furthermore, several participants thought these tools could serve as hands-on rainy day activities or as an extension after state testing is over.

One of the limiting factors preventing the broadening of participation in mathematics was teachers' content knowledge. This situation was partially due to the fact that concepts such as the Fibonacci sequence, golden ratio, and Voronoi patterns are not conventionally taught in K–12 education. However, several participants used deficit speech to describe their own mathematical knowledge. For example, Lucas noted, "I was good at math until

they started putting letters in there" and Mia explained that the content "was way out of my comfort zone, it's been 30 years since I have heard many of these terms". Yet, despite the lack of comfort with the advanced topics, Mia also saw value in the rectangular puzzle (workshop 1, activity 2) by noting that "I want to use the rectangular puzzle to introduce quadrilaterals, my [second grade] students are just learning that quadrilaterals have four sides". The use of these hands-on math tools also helped to uncover several misconceptions, including William stating that the 36-36-108 degree triangles in the decahedron (workshop 1, activity 1) could be used to teach the Pythagorean theorem despite the lack of right angles. Additionally, several participants placed their golden ratio spirals (workshop 2, activity 1) backward on pictures of snail shells and plants and did not seem to realize that the spirals did not align.

### 4.3.3. How Teachers Shared and Used the Tools

Results from the final survey and email correspondences showed that all nine teachers shared the tools and pedagogies with their family members, friends, students, and/or colleagues; implemented the activities in the classrooms; or made new digitally fabricated math tools (see Table 3). Eight participants shared the tools with family members. Five teachers showed the tools to friends who are not teachers (i.e., "non-education friends"). Five participants shared the tools with their students. Moreover, eight teachers introduced the concepts and tools and provided coaching feedback to their colleagues on technology integration using digital fabrication. For example, Mia wrote, "I demonstrated the Fibonacci plexiglass and golden ratio calipers to a colleague teaching a unit on da Vinci. She was excited to show those to her students".

**Table 3.** Frequencies of in-service teachers' usage of tools after the workshops (*n* = 9).

| Laser-Cut Math Tool | Students | | Colleagues | | Family | | Non-Ed Friends | |
|---|---|---|---|---|---|---|---|---|
| | *n* | % | *n* | % | *n* | % | *n* | % |
| Decahedron Tiles | **4** | **44** | **4** | **44** | **5** | **56** | 2 | 22 |
| Rectangular Puzzle | **4** | **44** | **5** | **56** | **5** | **56** | 2 | 22 |
| Voronoi Flipbook | 1 | 11 | 3 | 33 | 3 | 33 | 0 | 0 |
| Voronoi Stained Glass | 1 | 11 | 3 | 33 | **6** | **67** | 3 | 33 |
| Fibonacci Transparent Plexiglass | 0 | 0 | **5** | **56** | **5** | **56** | 2 | 22 |
| Golden Ratio Calipers | 2 | 22 | **6** | **67** | **5** | **56** | 3 | 33 |
| Origami Chompers | 0 | 0 | 2 | 22 | **4** | **44** | 2 | 22 |
| Icosahedron | 1 | 11 | 2 | 22 | **4** | **44** | 0 | 0 |
| String Art | 2 | 22 | 2 | 22 | **4** | **44** | 3 | 33 |
| Conic Sections | 3 | 33 | **5** | **56** | 3 | 33 | 3 | 33 |
| Nautilus Gears | 3 | 33 | 3 | 33 | 3 | 33 | 1 | 11 |
| Mercator Projections | 0 | 0 | 3 | 33 | 1 | 11 | 0 | 0 |
| Objects of Constant Width | 1 | 11 | 2 | 22 | 2 | 22 | 0 | 0 |
| Grand Total (*n* = 117) | 22 | 19 | 45 | 38 | **53** | **45** | 21 | 18 |

Note: Bold numbers show the tools that were shared by more than 40% of the participants.

Six teachers implemented some workshop math activities in their classrooms. At the elementary level, Mia reported, "I used the decahedron tiles with my 2nd-grade students when we were learning about polygons". Mia even created a new set of digitally fabricated math tools for implementation: "I made a rectangular puzzle out of cardstock for each of my students (I got the pattern from the MathHappens website) to play with when we were learning about quadrilaterals". Similarly, Ava let her 3rd-grade students use the decahedron tiles to make a circle and describe the patterns they noticed within the shapes. She also gave the rectangular puzzles to her students as a fun morning activity. Similar to Mia's classroom, the students were so intrigued by the activity, and they worked together to make more paper copies of the rectangular puzzles, so each student had a set they could manipulate. Additionally, while conic sections are usually not introduced until high school,

Riley noted, "For the conic sections, I used it with the 5th-grade math class I am working with to just show shapes".

Several of the activities were also popular at the secondary level; James described that some of his 7th-grade math students, "were interested in seeing how the objects of constant width worked as I described the task with placing the book on top and moving it around. The students then created their own objects of constant width using the 3D printer within the school". William noted that the laser-cut rectangular puzzle and decahedron tiles were an easy way for his 8th-grade math students to "begin exploring and thinking during free time". Chloe implemented the activities in several ways with her 9th-grade geometry students: "I asked students to create art using the square and strings and used the conic section to summarize the 3D to 2D cross-sections of the cone before our unit 4B test", and "I let my students play with nautilus gears as a fidgeting device".

### 4.3.4. Recommendations for Future Workshops

Participants shared their recommendations for future workshops during the semi-structured group interview. Suggestions included more technological knowledge, sustainability of digital fabrication, deep dive into the conceptual understanding of math content knowledge, workshops for different grade levels, readily available curricula, and interdisciplinary projects. Most participants suggested holding more technical workshops on digital fabrication in the future. They would like to see the behind-the-scenes steps of designing and digitally fabricating the math tools. They also want to learn how to design and create laser-cut files and how to produce their math tools for their classrooms. A few participants asked for workshop topics to be centered on the sustainability of digital fabrication in the K–12 schools, especially the cost-effectiveness of digital fabrication tools, cheaper options for the laser cutters, and alternative digital fabrication tools, such as Cricut machines. Almost all participants proposed that future workshops should be structured with a deep dive into one mathematics topic first, fully explaining the conceptual understanding of the mathematics content knowledge. Then, the workshop instructors should spend ample time on the maker activities and the discussion on how to transition from teaching the concept to adding more maker math activities. These teachers also favored the idea of designing workshops for different grade levels (elementary, middle, and high schools), so the math content knowledge could be fully aligned to the standards. All participants expressed that at the end of the workshops, they would like to have readily available lesson/unit plans, which could be taken back to the classrooms. In addition, some participants recommended future workshops should focus on technology integration into other content areas or interdisciplinary lesson/project ideas.

## 5. Discussion

Based on these findings, the following section discusses issues related to the misalignment of standards with goals of broadening the purposes of mathematics, PD suggestions for how math teachers can better integrate technology, and implications for future research and practice.

### 5.1. Misalignment between Standards and Broadening the Purposes of Learning Mathematics

These workshops sought to broaden the purposes of learning mathematics by exposing in-service teachers to the intersection of mathematical inquiry, nature, and the arts. Our findings indicate that teachers need a great deal more PD to shift their mindset beyond current standards-based practices. Based on the semi-structured group discussion, participants indicated that there were numerous systemic structures, policies, and practices in K–12 education that continue to undermine their ability to teach for deep mathematical understanding. Currently, there is a misalignment between state math standards and the broader goal of teaching mathematics as a coherent whole [4].

In order to broaden the purposes of learning mathematics, teachers and teacher educators need to address the tension that exists between the procedural versus conceptual

understanding of mathematics. There is a difference between being able to "do" the math procedures associated with functions versus understanding their real-world applications. For example, students learning linear functions often solve for slope and intercept without enough context to develop a deep understanding of what those values mean. Instead, mathematical inquiry of the natural world provides students with a context that is more likely to spur connections that may lead to an appreciation for the wonder, joy, and beauty of mathematics. However, it is important to note that despite the numerous connections between nature and mathematics, the word "nature" does not appear in most state standards or the common core outside of the "nature of attributes" in data and "quantities in nature and society" [84]. Furthermore, the Fibonacci sequence is only mentioned once in common core as an example of how to define a function recursively.

To address this misalignment, teachers should look to nature to find hands-on activities where students can engage in mathematical inquiry. For example, concepts such as pi are usually taught with rote memorization to the nearest hundredth or provided to students on a formula sheet. Instead, students can "discover" pi by going outside and hugging a tree with a rope, measuring the diameter of the trunk, and comparing their answers with their peers to socially construct its numerical value. Similarly, digitally fabricated math tools can support the development of deeper learning by allowing students to search for naturally occurring golden ratios with the use of clear acrylic spirals and golden ratio calipers. Furthermore, 3D-printed tools, such as the Mercator projection, can support interdisciplinary investigations into the nature of light, the curvature of the world, and how visual representations can be used to distort the truth.

While the goal of broadening the purposes of learning mathematics deserves praise, the idea continues to face the challenges in a standardized world. Wonder, joy, and beauty are concepts that are very difficult to standardize and measure. Since beauty is in the eye of the beholder, teachers need more opportunities to move beyond the standards and develop an appreciation for mathematics within the context of the natural world. Teacher education and PD programs may choose to address this issue through mathematical field trips to local botanical gardens and museums to discover how math is embedded in nature and the arts. Furthermore, teacher education programs may consider adding a course focused on math appreciation to help teachers reconceptualize mathematics beyond number and symbol manipulation.

### 5.2. Professional Development for Math Teachers to Integrate Technology

Both the mathematics concepts and technology tools chosen for the workshops were somewhat unfamiliar and uncomfortable for the participants, as evidenced in the field notes, semi-structured interviews, and follow-up survey. Results from the data indicated that participants tended to overfocus on the technical use of the technology tools [79,85] and connections to content standards instead of the higher-order thinking skills.

Participants in this study also encounter some similar barriers to implementation as those identified in the literature [79]: (a) teachers' beliefs that technology-rich activities of higher-order thinking skills are time-consuming, ineffective, and needy of too much preparation [86] and (b) school culture being relatively conservative and not open to integrating more technology-rich tasks in the classrooms [87]. As a result, similar to a study conducted by Bray and Tangney [77], this study also found a misalignment between the research-based best practices for math technology integration advocated in the PD workshops and teachers' actual implementation in their classrooms. The results of the final survey illustrated that except for one teacher (11%), the other four teachers (44%) who implemented the activities in their classrooms did not design technology-rich math activities to address students' higher-order thinking skills, not to mention the four other teachers (44%) who did not implement any activity in their classrooms after the workshops.

This study also found that school culture played a major role in whether the technology would be implemented following PD. Issues such as misalignment with the mission and vision of the school and district along with administrative priorities were factors that

impeded teachers from using these tools in their practice. Those teachers that did envision implementing tools from the workshops tended to focus on gifted and talented populations, indicating that high-tech math tools have the potential to replicate the Matthew Effect, where the technology-rich become richer; while the poor become poorer [88]. In order to address issues related to school culture, those creating PD sessions should think about including a broad range of stakeholders including teachers, administrators, parents, and community members who have outside expertise. This may help build capacity within the organization to create buy-in from school administration and the broader community, which can help shape the mission and vision of the school and district to support technology-rich learning environments. Those leading workshops should consider open discussion with teachers about what the school culture means within the context of their implementation and provide teachers with strategies to help them overcome barriers.

### 5.3. Added Value of the Present Work

This study is situated within the context of multiple initiatives that seek to broaden the study of mathematics in relation to nature and the arts. While the guiding premise of the study stems from the NCTM's *Catalyzing Change* series [1–3], the authors acknowledge that international efforts are taking place within the global FabLab network to explore the intersection of mathematics, nature, and the arts through digital fabrication [45]. Integrating new tools, technologies, and pedagogies is challenging since school systems and teacher education programs do not teach integrated curricula. This is partially due to the limited amount of time teachers have to teach the established curricula and the teaching dilemma that emerges from the tension between agency and structure in the classroom [89]. However, the maker projects outlined in this study are very simple from the perspective of design and fabrication and only require basic computer-aided design (CAD) and laser-cutter skills to create any of these mathematic manipulatives.

Findings from this study add value by sharing numerous digitally fabricated mathematic manipulatives in the context of teacher PD workshops. While these workshops took place at a maker lab that had access to a consumer-grade laser cutter and 3D printer, the study also highlighted the ingenuity of several participants to use conventional tools in their classroom to make class sets of the rectangular puzzle and decahedron tile activities. In the hopes of minimizing the digital divide, future studies are planned that will examine how teachers can design their own mathematic manipulatives using CAD and vector applications. Rather than relying entirely on 1/8" plywood and acrylic, these future workshops will also explore inexpensive materials such as cardboard and cardstock and alternative digital fabrication tools such as the Silhouette Cameo and Cricut machines, which are more affordable for most schools.

### 5.4. Implications for Future Research and Practice

The findings of the current study reiterated that there exists a misalignment between what is taught in the PD workshops and teachers' implementation. Thus, researchers and teacher educators need to further investigate PD workshops' learning environment, topic, design, and length as well as school cultures in order to improve teachers' TPACK development and self-efficacy in technology integration [90]. As Polly and Hannafin summarized and promoted, PD programs focusing on technology integration should follow the learner-centered principles, which are student-focused, teacher-owned, collaborative, content and theory-laden, reflective, and comprehensive [86]. Future research could use these principles to guide the design of the PD workshops and test their effectiveness on in-service teachers' development of TPACK and other relevant variables. Some researchers also suggested that PD programs for technology integration in mathematics classrooms should emphasize conceptual understanding, teaching strategies, time, assessment, grade level, and student engagement [69,71,74]. Each of these variables could be examined further in future studies. Additionally, instruments examining teachers' attitudes and

self-efficacy could be used to triangulate data to better understand the effectiveness of the PD programs [91].

For future practice, the researchers recommend the following suggestions for PD workshop design. First, future PD workshops should be designed specifically for different grade levels (elementary, middle, and high school) so that each workshop can focus on specific mathematics standards/goals and provide teachers with opportunities to consider how to implement the technology-rich math tasks with their students [92]. Workshop instructors should allocate ample time for teachers to reflect and discuss how these technology tools can support student learning during math tasks. It is worth investigating whether teachers increase their TPACK development in the workshops designed for their grade levels.

Second, PD workshops should be designed to be more standards-focused and student learning-outcomes-focused to potentially address the challenge of lack of time for implementation [86,93]. Teacher educators could provide practitioners with opportunities to explore high-level mathematical tasks in which digitally fabricated tools support their learning using a practice-based approach [94–96]. In this approach, in-service teachers will first explore mathematical tasks as learners. After fully unpacking the mathematics for themselves, teachers can consider how best to support and facilitate students' learning.

Third, more time and exposure will be ideal to make the PD ongoing and comprehensive, so teachers will have more support on math conceptual understanding and domains in TPACK. The researchers of the current study plan to offer another set of PD workshops focusing on how to design digital fabrication tools using the software and mass produce the designs for teachers' classrooms. In addition, it will be optimal to go to schools and classrooms to provide additional coaching for implementation [97–99]. Longitudinal research investigating how to support ongoing PD is greatly needed.

Lastly, PD workshops should consider school culture, including leadership styles and administrative barriers [98,100,101]. Teacher educators should provide teachers with opportunities to develop and practice strategies that promote innovation within their schools [102]. In-depth research on the relationship between school culture, PD, and innovation will contribute to PD policy and practice.

## 6. Limitations

Limitations of this study include using recruiting participants through convenience sampling of teachers who were enrolled in an instructional technology graduate program. Due to this sampling method, some participants may already have a bias toward believing in the benefits of technology in education. All participants were enrolled at one university in the southeastern United States, which means the findings of this study may not represent the mathematical culture of other regions or countries. Additionally, this study took a "buckshot" approach by introducing a large number of tools rather than focusing on how individual tools could be applied in grade-specific contexts. While the follow-up survey captured how several of the teachers implemented the tools in the classroom, this study was not able to directly measure student reactions to these tools. Additionally, this study took place in February and March of 2022, while COVID protocols were still active. As such, it was not possible to recruit enough participants to conduct this study with a control group using more conventional mathematics manipulatives. Lastly, participants had limited access to digital fabrication tools, with only one laser cutter and one 3D printer being accessible in the maker lab. As a result, researchers produced the majority of tools in advance, which provided fewer opportunities for participants to develop their technology knowledge and learn technical skills, such as setting the power and speed of the laser cutter for different materials (e.g., plywood, cardboard, and acrylic).

## 7. Conclusions

This study aimed to examine how digitally fabricated mathematics tools can be used to help in-service teachers explore the wonder, joy, and beauty of mathematics through the authentic connections that exist between mathematics, nature, and the arts. Findings

from the study indicate that digitally fabricated math tools need to be aligned with grade level, content, and standards to be successfully adopted by teachers. Furthermore, the affordances of digitally fabricated math tools included opportunities for hands-on learning, visual support at the secondary level, and real-world connections that go beyond the state standards. Barriers include purchasing a laser cutter, ventilation and noise issues, time constraints (i.e., too many meetings, no time for training, preparation time to create materials), misalignment with school and district mission and vision, and a lack of administrative support.

Based on the goal of broadening the purposes of learning mathematics, this study found that there is a misalignment between the current standardized mathematics curriculum and the broader goals of appreciating the wonder, joy, and beauty of mathematics. Evidence from participants suggests that some of the math topics included in these workshops (e.g., Fibonacci sequence, golden ratio, and Voronoi diagrams) can be used to help teachers form a contextual understanding of the connections between nature and mathematics. Low-tech math tools, such as laser-cut wooden geometry tiles, were more accessible and could serve as an entry point for teachers interested in introducing new math manipulatives to their classrooms and curricula. More research is needed to better understand how PD can support teachers in the development of their TPACK through the design, creation, and implementation of digitally fabricated math resources.

**Author Contributions:** Conceptualization, J.R.H., Y.J. and A.H.; methodology, J.R.H. and Y.J.; formal analysis, J.R.H., Y.J. and A.H.; investigation, J.R.H.; resources, J.R.H. and L.S.; data curation, Y.J., A.H. and L.M.; writing—original draft preparation, J.R.H., Y.J. and L.M.; writing—review and editing, J.R.H., Y.J., A.H., L.M. and L.S.; visualization, J.R.H., Y.J. and L.S.; project administration, J.R.H.; funding acquisition, J.R.H. All authors have read and agreed to the published version of the manuscript.

**Funding:** This research was funded by the School of Instructional Technology and Innovation at Kennesaw State University.

**Institutional Review Board Statement:** The study was conducted in accordance with the Declaration of Helsinki, and approved by the Institutional Review Board (or Ethics Committee) of Kennesaw State University (protocol code IRB-FY22-339, approved 10 February 2022).

**Informed Consent Statement:** Informed consent was obtained from all subjects involved in the study.

**Data Availability Statement:** The data presented in this study are available on request from the corresponding author. The data are not publicly available to maintain the privacy of participants. Templates of tools can be found at https://mathhappens.org.

**Acknowledgments:** The authors would like to acknowledge Lauren Siegel and the MathHappens Foundation for designing, developing, and donating many of the laser-cut math tools that were used in these workshops.

**Conflicts of Interest:** The authors declare no conflict of interest.

## Appendix A

**Table A1.** Description and photos of workshop activities.

| Workshops | Activities | Examples |
|---|---|---|
| Workshop 1 | **1. Decahedron tiles (20 min)**<br>• Teachers first freely play with the pieces without instructions.<br>• Then, teachers discuss their observations.<br>• Next, teachers need to use all 20 triangle pieces to form a "round" shape.<br>• Teachers are given instructional sheets with five unique designs.<br>• The instructors lead a discussion of the activity focusing on shapes, colors, and angles. | 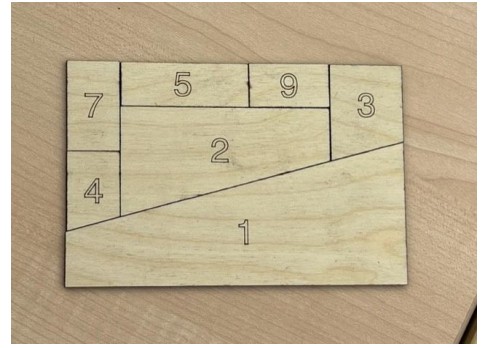 |
| | **2. Laser-cut rectangular puzzle (20 min)**<br>• Participants received the packs of pieces and peeled off the blue painter's tape used to prevent smoke marks on the laser-cut pieces.<br>• Then, they begin the challenge to figure out how to make larger rectangles with 5, 6, 7, and 8 pieces.<br>• The instructors lead a discussion on the activity and how it can be implemented in the curricula of various grade levels. | |
| Workshop 1 (cont.) | **3. Voronoi flipbook (30 min)**<br>• The instructors first lead a discussion on nature and Voronoi patterns and then send handouts to teachers to trace different Voronoi patterns.<br>• Everyone discusses their observations of the patterns.<br>• The instructors lead an unplugged activity where participants stand up and space themselves to create a Voronoi pattern.<br>• Next, the instructors introduced a simulation: the football players' position data with animated Voronoi charts in R.<br>• Teachers work on cutting and creating their Voronoi flipbooks and explore how the shapes change when they collide with each other. | 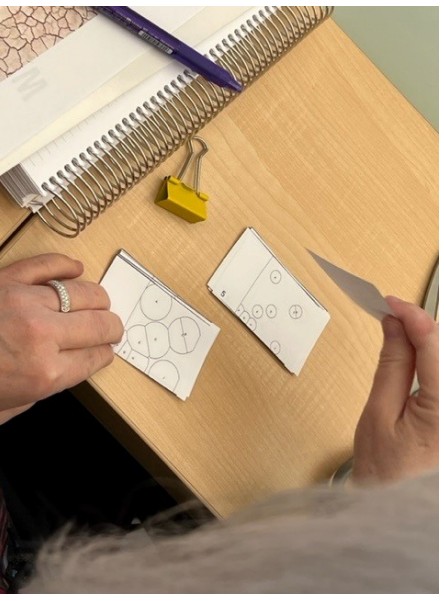 |

**Table A1.** *Cont.*

| Workshops | Activities | Examples |
|---|---|---|
| | 4. **Voronoi stained glass (45 min)** <br> • Participants begin tracing the frames on the tracing paper once receiving all materials. <br> • Then, they cut the tracing paper according to the shape of the frame and design their own Voronoi patterns that are meaningful to them. <br> • The instructors talk about the math content knowledge of the Voronoi patterns. <br> • Participants color their designs and glue the tracing paper and frames together. <br> • The instructors lead a discussion on how to use this activity in the classrooms at various grade levels. | 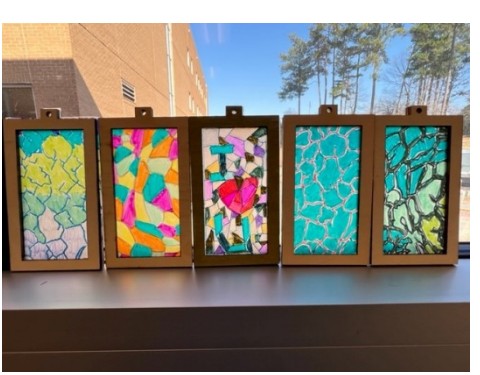 |
| Workshop 2 | 1. **Fibonacci transparent plexiglass (40 min)** <br> • The instructors lead a brief discussion of the golden ratio and hand out grid paper for teachers to graph the Fibonacci sequence. <br> • Teachers draw their graphs following the sequence 0, 1, 1, 2, 3, 5, 8, 13, 21, 34, 55 and write down the ratio of the shape on the top of the paper. Some teachers collaborate. <br> • The instructors hand out the laser-cut Fibonacci transparent plastics. <br> • Teachers peel off the protective coating and use the acrylic pieces to find golden ratios in books, displays, nature, and artworks while discussing with each other and the whole group. | 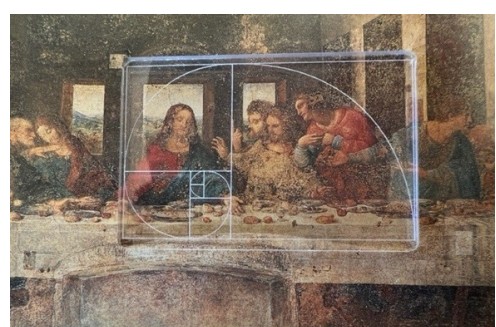 |
| | 2. **Golden ratio calipers (50 min)** <br> • The instructors hand out the golden ratio sheet and explain the origins of the golden ratio and the Fibonacci sequence. <br> • Then, the whole group discusses a few examples. <br> • The instructors pass around a book with Renaissance art and demonstrate how to find golden ratios in the art using the golden ratio calipers. <br> • Then, the instructors explain how to assemble the calipers and hand out the materials, four laser-cut pieces, four bolts, four nuts, four washers, and two wrenches. <br> • Teachers pick up the supplies and assemble the calipers. Peer groups work together. <br> • After, teachers measure their bodies and other items to find golden ratios in real life. <br> • The whole group reflects on this activity and shares their findings. | 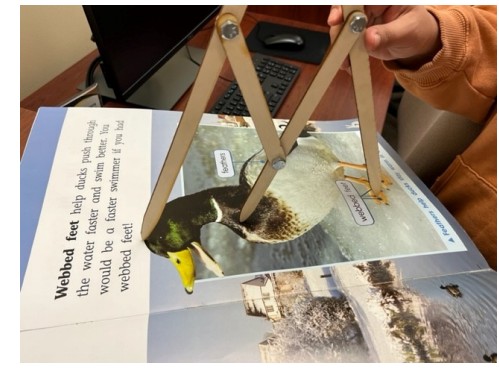 |

**Table A1.** *Cont.*

| Workshops | Activities | Examples |
| --- | --- | --- |
| | **3.**    **Origami chompers (10 min)** <br> • The instructors hand out pre-cut chompers, googly eyes, and glue sticks and explain the design and materials of this activity. <br> • Teachers fold and decorate their chompers. <br> • Then, the instructors lead a whole group discussion on the use of the chompers, the design of origami, and implementation ideas. | 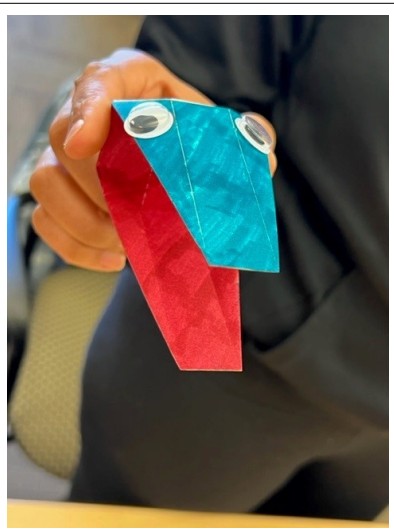 |
| Workshop 2 (cont.) | **4.**    **Icosahedron (20 min)** <br> • The instructors hand out pre-packaged sets to teachers. <br> • Teachers work on assembling the icosahedron. <br> • Then, the whole group discusses how many triangles and rubber bands the icosahedrons use. Everyone continues to discuss how this activity can be used in the classrooms in various content areas and grade levels. | 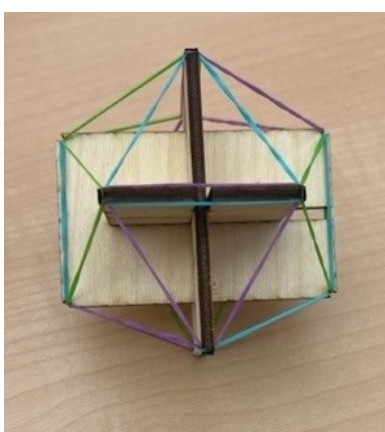 |
| | **5.**    **String art (30 min)** <br> • The instructors hand out materials to teachers, the laser-cut frames and strings. <br> • Teachers select the shape of the frames and the colors of the strings. They cut the strings and work on creating their own patterns. <br> • Teachers walk around the maker lab to find inspiration for the string art. Some teachers work on multiple frames during the activity. <br> • Then, the whole group discusses how this activity can be used in the classrooms in various content areas and grade levels. | 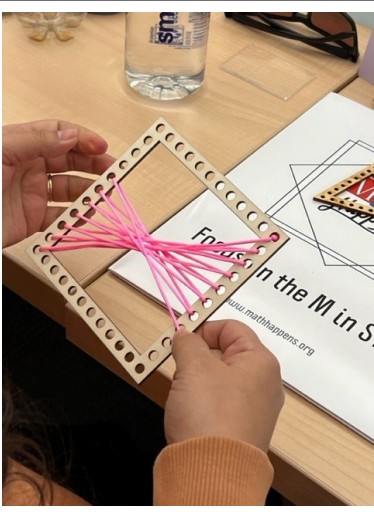 |

**Table A1.** *Cont.*

| Workshops | Activities | Examples |
|---|---|---|
| Workshop 3 | **1. <u>Conic sections</u> (20 min)**<br>• The instructors hand out the laser-cut conic section pieces and playdoh.<br>• Teachers first assemble the conic sections and then work on forming the playdoh into cones. Then, they use rulers to cut the playdoh imitating the conic section models.<br>• Then, the whole group discusses how this activity can be used in the classrooms at various grade levels. | 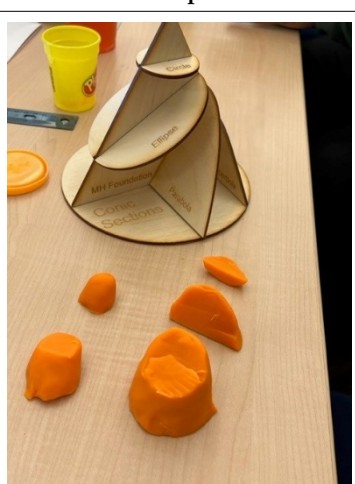 |
| | **2. <u>Nautilus gears</u> (20 min)**<br>• The instructors showed the pre-3D-printed template for Nautilus gears and discuss the material used for the 3D printing.<br>• Teachers get their templates and peel off the individual parts for the gears.<br>• Teachers drill the holes to make them bigger.<br>• Teachers assemble the gears in teams.<br>• Then, the whole group discusses how this activity can be used in the classrooms at various grade levels. | 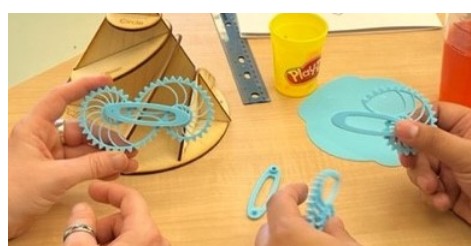 |
| | **3. <u>Mercator projection</u> (20 min)**<br>• The instructors hand out the pre-3D-printed objects to teachers.<br>• Teachers experiment by shining their cell phone light through the objects and observing the changes in the shapes of the shadows.<br>• The instructors explain the phenomenon and the concept of Mercator projection.<br>• Then, everyone goes to the website <u>thetruesize.com</u> using their laptops and phones. The group compares the sizes of different countries and sees how the Mercator projection was adopted in map-making.<br>• Then, the whole group discusses how other content areas such as social studies can be taught with math content knowledge. | 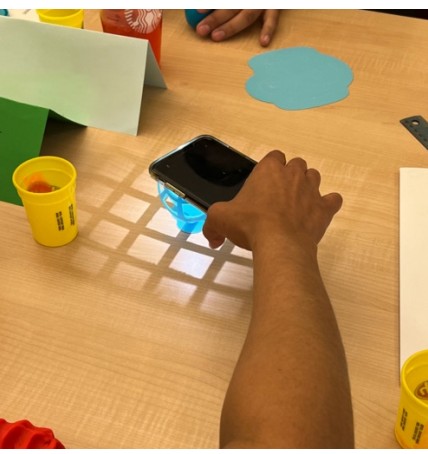 |

**Table A1.** *Cont.*

| Workshops | Activities | Examples |
|---|---|---|
| Workshop 3 (cont.) | **4.** **Objects of constant width (35 min)**<br>• The instructors hand out the pre-3D-printed objects to teachers and let them experiment.<br>• Teachers conduct a variety of experiments, for example, putting the shapes under a thick book and then rotating the book.<br>• Teachers sit together on the carpet to do more experiments and share their thoughts with each other.<br>• The instructors showcase how to laser cut cardboard into the foundational shape of this object of constant width and explain in detail how this shape forms.<br>• Then, the whole group discusses how this activity can be used in the classrooms at various grade levels. | 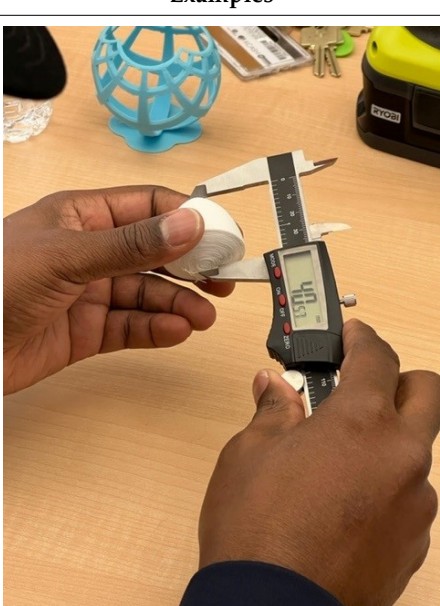 |

**Appendix B**

**Semi-structured group interview questions:**

1. What are some affordances and constraints of the laser-cut math tools in your perspective?
2. What are some of your ideas about how these laser-cut math tools and activities could be implemented in your classrooms?
3. What challenges do you have if you want to implement these laser-cut math tools in your math curricula? What viable solutions do you see for these challenges?
4. What connections between nature and mathematics came up for you during the workshops? Between the workshops?
5. What are some of your suggestions and recommendations for future workshops? How can we improve and better serve your needs?

**Appendix C**

**Follow-up survey:**

What is your first and last name?

Which tools have you used with students, colleagues, family, and/or non-educational friends? Check all that apply. (Please see e-mail for link to photos of tools).

| Maker Tools | Students | Colleagues | Family | Non-Education Friends |
|---|---|---|---|---|
| Decahedron tiles | | | | |
| Laser-cut rectangular puzzle | | | | |
| Voronoi flipbook | | | | |
| Voronoi stained glass | | | | |
| Fibonacci transparent plexiglass | | | | |
| Golden ratio calipers | | | | |
| Origami chompers | | | | |
| Icosahedron (rubber band shape) | | | | |
| String art | | | | |
| Nautilus gears | | | | |
| Mercader projections | | | | |
| Objects of constant width (the spheroids) | | | | |

Please provide a brief description about how you have used any of the tools you have checked above.

Is there anything else you would like to share with us?

**Appendix D**

**Table A2.** Sample of the qualitative codebook for the decahedron and rectangular puzzle activities.

| Activity | Technological Knowledge | Pedagogical Knowledge | Content Knowledge | Contextual Knowledge | Aesthetics | Participant Examples | Misconceptions | Additional Notes |
|---|---|---|---|---|---|---|---|---|
| Decahedron (triangular pieces) | 3 mm plywood cutting speed/power, how to color plywood with water-based bingo dabbers. | Discovery learning/free play. | Teachers mention their observation of the triangular pieces are similar to Tangrams. | The Instructor offered the opportunity to get the instructional sheet that has five designs on it for the activity. Four teachers decided to take the sheet and explored the patterns, and the other four teachers decided to continue to try themselves. | After successfully coming up with new patterns, teachers immediately engage in conversations about how the shapes look like in the patterns and how colors play into the design. "Does that shape remind you of anything?" Teachers also want to make the patterns visually pleasing by coordinating the colored pieces. | Stop sign, decagon, paper airplane, star. | Connection to Pythagorean theorem (Triangular pieces have no right-angle, 36-36-108 degrees). | Participants were asked to use all 20 pieces to make a "round" shape after a period of free play. Participants were informed to not use the hexagonal piece from their baggie of pieces. |
| | | | The instructor asked the teachers the degrees of the triangles and asked them to think about their degrees. "How many sides are there in the pattern?" "10 sides, 1440-degrees (sum of the interior angles, (n-2)*180. (NOTE: Participants identified the interior with 36*10 = 360 degrees). | Instructor asked the kindergarten teacher what she thinks about this activity for kindergarteners, participants thought the activity was good for all ages. | Teachers commented that they began playing by grouping the pieces based on colors, and then, it came together into the full pattern. Most teachers agreed that colored pieces are more appealing than plain wood pieces, and the colors help with the play and design. | | | NOTE: Not mentioned during workshop, this is the Phi triangle, with phi as the short legs, and hypotenuse of phi + 1. Also called the Divine Triangle. See video: https://www.youtube.com/watch?v=z4hCcI_Ates(accessed on 19 August 2022) |
| Rectangular Puzzle | Activity packs were distributed to teachers, and teachers peeled off the stickers (blue painter's tape) on the pieces. Teachers mentioned they smelled like a campfire. | Challenge-based learning. | Elementary teachers, similar to manipulatives. Middle school teachers, surface areas, and columns. High school teacher, very different from high school math teaching, rational expression, logs, imaging numbers. Everyone agreed that we were doing geometry. | Challenge: Find a rectangle with 5, 6, 7, and 8 pieces. | Some people liked the colors; one said color did not make any difference, and some preferred plain wood color. | (Real-world example) Similar to buildings and lands, this can be used by people doing architecture and lands. | 6 vs. 9 for upside-down piece of puzzle due to symmetrical piece. | "The challenge is to try to figure out how to make large rectangles and how to make the rectangles with 5, 6, 7, and 8 pieces". One group figured it out immediately and said, "Let's just do the eight pieces really quick". Teachers celebrated once they completed a rectangle. Everyone was very engaged, and there was a lot of laughter. |

**Table A2.** *Cont.*

| Activity | Technological Knowledge | Pedagogical Knowledge | Content Knowledge | Contextual Knowledge | Aesthetics | Participant Examples | Misconceptions | Additional Notes |
|---|---|---|---|---|---|---|---|---|
| | One teacher asked about the 3D shapes. Whether we can print the 3D shapes for these activities? The instructor showcased the 3D printed Pi vase to teachers, 28 h of printing. | How this activity will be easier or harder if we don't have the numbers on it?" One teacher mentioned that it might be easier. | | | | One teacher talked about city planning, figuring out the roads, parallels, different building shapes, and so on. | | The instructor showcased the book, *Earnest Irving Freese's Geometric Transformations: The Man, the Manuscript, the Magnificent Dissections!* written by Greg N. Frederickson and talked about 15 ways to do pentagons and talked about tiles and quilting. |

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
