# Peer review of "Maker Math: Exploring Mathematics through Digitally Fabricated Tools with K–12 In-Service Teachers"

_mathematics, doi:10.3390/math10173069_

Round 1

Reviewer 1 Report

This study aims to examine how digitally fabricated mathematics tools can be used to help in-service teachers explore the wonder, joy, and beauty of mathematics through the authentic connections that exist between mathematics, nature, and the arts.

This is an important topic; the research questions are well defined as well as the methods and analysis. The only weakness that I identified: I am curious to read more on how students reacted to the introduction of the new tools used by teachers in the classroom. I believe it will enhance the paper to add a dedicated paragraph to section 4.3.3, summarizing students' reaction (and if possible, comparing a pilot vs test group's reactions). 

Author Response

Dear editor and reviewers:

Thank you so much for your feedback on our manuscript. We have revised the paper according to your suggestions and provided detailed explanations in the tables below. We appreciate your support and help to make our paper stronger.

Point 1: This is an important topic; the research questions are well defined as well as the methods and analysis.

Response 1: The authors would like to thank you for the compliment about our methodology and analysis.

Point 2: The only weakness that I identified: I am curious to read more on how students reacted to the introduction of the new tools used by teachers in the classroom. I believe it will enhance the paper to add a dedicated paragraph to section 4.3.3, summarizing students' reaction (and if possible, comparing a pilot vs test group's reactions). 

Response 2: Data on how students reacted to the tools in the K-12 classroom are limited to post-workshop survey data from the teacher participants since our IRB did not include the use of participants under the age of 18. I have revisited all survey data to expand on how teachers used the tools with their students. Using this data, I have expanded upon section 4.3.3 and attempted to include student reactions whenever possible and separated the findings into elementary and secondary grade levels. The lack of direct measurement of students’ reactions to the tools has been added to the limitations. Please see the highlighted texts in these two sections.

As for a comparison between pilot and test group reactions, many of these tools have been implemented as part of workshops with the MathHappens Foundation for almost a decade. The lead author has been involved with this organization for many years, but this is the first formal IRB-approved research study done with these tools. We look forward to using the current dataset and findings to inform a follow-up study that we are planning for the next academic year. Thank you for pointing out future directions for us.

Reviewer 2 Report

Summary of the Work

The main objective of this work is to examine how digitally fabricated mathematics tools can be fruitfully used by teachers for exploring mathematics through its connection between mathematics, nature, and art.

Main Results Obtained

- There is a misalignment between the current standardised mathematics curriculum and the broader goals of appreciating mathematics in all of its aspects.

- Some of the math topics included in these workshops may help teachers to establish connections between nature and mathematics.

- Low-tech math tools, such as laser-cut wooden geometry tiles, are more accessible and could help teachers interested in introducing Fab-Lab maths to their classrooms and curricula.

General Remarks 

- This work belongs to a wide series of works, initiatives, and workshops that have recently appeared also in literature, aimed at introducing the Fab-Lab method in primary and secondary schools to explore mathematical sciences in a broader and more harmonious context that ranges from nature to art (e.g., Fablab Nordvest, Fab-Lab Maths and Art, introduced by Prof. Pierre Coullet - Université Côte d'Azur (France) (this method is in an experimental phase and is currently adopted by some secondary schools in Santiago de Chile under the form of a series of workshops), Fab Lab Kambsdalur, etc.)

- It is customary to specify the acronyms when they appear for the first time in the manuscript even when they are well-known in the literature (e.g., please specify the acronym TPACK in the abstract etc.).

- The terminology K-12 education and educational technology is used in the United States, Canada, and in other, but not in all, countries (e.g., not in Europe). For easy reference, it is advisable to insert a footnote where it is recalled that K-12 is a short form for the publicly-supported school grades prior to college. These grades are kindergarten (K) and the 1st through the 12th grade (1-12).

- The work is interesting and its objectives are well illustrated. However, there are some points that need to be clarified. In particular, is not evident to identify the added value, or the innovative aspects, brought by this work compared to similar initiatives proposed and currently being implemented. The following suggestions are aimed to clarify

(By the way, it is mentioned that the reviewer is currently in charge of officially working on educational programs at EU level. For this reason, he is very interested in this research/education subject).

Suggestions

As mentioned, the work is interesting and, in my opinion, initiatives of this type should be encouraged. However, I kindly ask that the authors answer concisely, motivating their answers, to the following series of questions that currently constitute the main drawbacks of this type of initiatives (from the STEAM education to the Fab-lab initiatives currently proposed in the literature).

1) Generally, the initiatives Fab-Lab do not provide clear-cut guidelines for educators to follow. In addition, most school systems do not teach integrated curriculums, therefore teachers do not know how to integrate. The authors are asked to explain how these drawbacks may be overcome.

2) The main objection addressed to the Fab-Lab methodology is that this method requires many hours of lessons and practice. This can come at the expense of other subject matter, such as history, philosophy, literature etc. In concrete terms, what do the authors propose to establish the hours of lessons dedicated to Fab-Lab methodology without compromising the other subject matters, equally fundamental for the education and training of the student? (It goes without saying that literature, philosophy, religions, history, civic education, etc. are also subjects of fundamental importance for the "forma-mentis" of the student).

3) One of the main problems that Fab-Lab proposers have had to face in secondary schools in Chile is that, actually, few elementary and secondary teachers are qualified to teach in the areas ranging from math, nature, to art. The same problem arises in the present work. How do the authors intend to tackle and solve this problem? By organising a series of training courses for teachers?

4) Apart from the technical problems, there are also problems of a purely administrative and legal nature to be solved. More specifically, Europe is actually facing the problem of "Unification and Harmonisation" of the study courses and diplomas. This should apply to the Countries of West Europe as well as the Countries of Eastern Europe. Now, if the Fab-Lab method were to be adopted at a European level to primary and secondary schools, there would be no "National standards" or "Teacher certification" in this case. What suggestions can the authors provide on this issue?

5) Last but not least, the economic aspect must also be considered. Indeed, the Fab-Lab methodologies can be costly (ref., for instance to Forside - fablab, FabLab Kambsdalur etc.). Many countries (for example several countries of Latin America and possibly other countries) are not able to support these expenses at national level, but only for some schools and only on an experimental basis. This would lead to a divergence at the educational level between "rich" and "less rich" countries. What is the opinion of the authors on this?

6) Finally, considering the answers to the above questions, can the authors insert a short paragraph specifying the added value of the present work compared to similar initiatives currently proposed and reported in the literature?

Conclusions

As said, the work is interesting and it was a pleasure to read it. In my opinion it deserves to be published. However, many works, initiatives, and workshops concerning the Fab Lab methodologies, based on digitally fabricated mathematics tools, have appeared in the literature (for example, in the EU there are several EC sub-programs entirely dedicated to this topic). In order to stimulate and attract the interest of the reader more, it is therefore advisable, in my opinion, to show the added value brought by this work compared to similar and existing initiatives, maybe with the help of the answers to the questions raised above.

Author Response

Dear editor and reviewers:

Thank you so much for your feedback on our manuscript. We have revised the paper according to your suggestions and provided detailed explanations in the tables below. We appreciate your support and help to make our paper stronger.

General Remarks

Point 1: This work belongs to a wide series of works, initiatives, and workshops that have recently appeared also in literature, aimed at introducing the Fab-Lab method in primary and secondary schools to explore mathematical sciences in a broader and more harmonious context that ranges from nature to art (e.g., Fablab NordvestFab-Lab Maths and Art, introduced by Prof. Pierre Coullet - Université Côte d'Azur (France) (this method is in an experimental phase and is currently adopted by some secondary schools in Santiago de Chile under the form of a series of workshops), Fab Lab Kambsdalur, etc.)

Response 1: Thank for you the suggestion about aligning this work with the initiatives that are taking place with the Fab Lab networks, specifically in the European and South American contexts. The authors appreciate the suggested research and FabLabs. Based on a literature review, the authors were unable to locate any specific peer-reviewed or grey literature articles about these current initiatives, however, another study by Jeldes et al. with Aconcagua FabLab has been added to the end of section 2.1 to support the initiatives in other countries. Please see the highlighted text.

Point 2: - It is customary to specify the acronyms when they appear for the first time in the manuscript even when they are well-known in the literature (e.g., please specify the acronym TPACK in the abstract, etc.).

Response 2: Thank you for the suggestion. The abstract has been updated to spell out Technological Pedagogical and Content Knowledge (TPACK).

Point 3: - The terminology K-12 education and educational technology is used in the United States, Canada, and in other, but not in all, countries (e.g., not in Europe). For easy reference, it is advisable to insert a footnote where it is recalled that K-12 is a short form for the publicly-supported school grades prior to college. These grades are kindergarten (K) and the 1st through the 12th grade (1-12).

Response 3: A footnote has been added to the second mention of K-12 due to formatting issues with footnotes on the first page. Thank you for this suggestion.

Point 4: - The work is interesting, and its objectives are well illustrated. However, there are some points that need to be clarified. In particular, it is not evident to identify the added value, or the innovative aspects, brought by this work compared to similar initiatives proposed and currently being implemented. The following suggestions are aimed to clarify.

(By the way, it is mentioned that the reviewer is currently in charge of officially working on educational programs at the EU level. For this reason, he is very interested in this research/education subject).

Response 4: Thank you for this suggestion. In our context, publicly-supported teachers have not had the opportunities to experience laser-cut math tools and learn about how to integrate those into their classrooms. The innovative aspect of our paper is to provide professional learning to the elementary, middle, and high school teachers by using computer-aided design and laser-cutter to create math tools for their classrooms, which is both a technology and pedagogy innovation for these teachers and their schools. This type of study has not been published in the literature, so the added value of our paper is twofold: (1) Provide empirical data of such a study to inform the field that this method may help teachers establish connections between nature and mathematics and explore the wonder, joy, and beauty of mathematics, and (2) Disseminate the methods we used for professional learning and help other teacher educators use similar approaches in their courses or workshops. We wrote a section 5.3 Added Value of the Work in the text to highlight the added values.

Sugeestions

Point 5: As mentioned, the work is interesting and, in my opinion, initiatives of this type should be encouraged. However, I kindly ask that the authors answer concisely, motivating their answers, to the following series of questions that currently constitute the main drawbacks of this type of initiatives (from the STEAM education to the Fab-lab initiatives currently proposed in the literature).

Response 5: Thank you for the suggestions. Please see our answers below.

Point 6: Generally, the initiatives Fab-Lab do not provide clear-cut guidelines for educators to follow. In addition, most school systems do not teach integrated curriculums, therefore teachers do not know how to integrate. The authors are asked to explain how these drawbacks may be overcome.

Response 6: The authors support Fab-Lab initiatives and recognize the global effort that it has taken to establish and sustain these spaces and communities. From the perspective of the authors, Fab-Labs are not the dominant paradigm in North America, largely due to the cost of standardization and hours of training. As such, we view this work through the lens of makerspace culture, which is much more informal and often utilizes equipment in school libraries and engineering labs.

We do not necessarily view this study as an integrated curriculum. From one perspective, this study could be viewed as integrating math and engineering. However, mathematics has a long tradition of creating mathematical manipulatives to help students visualize concepts in a tangible form. Based on the finding from this study and the interest of our participants, we have planned a follow-up study that will more closely examine the integration question with the teachers learning how to use CAD to design, produce, and implement tools in their classrooms. We plan to further examine the strengths of the integrated curricula and their impacts on teachers and students. We also anticipate this follow-up study will broaden our perspectives on how the drawbacks of integration can be addressed at the primary and secondary grade levels.

Point 7: The main objection addressed to the Fab-Lab methodology is that this method requires many hours of lessons and practice. This can come at the expense of other subject matter, such as history, philosophy, literature, etc. In concrete terms, what do the authors propose to establish the hours of lessons dedicated to Fab-Lab methodology without compromising the other subject matters, equally fundamental for the education and training of the student? (It goes without saying that literature, philosophy, religions, history, civic education, etc. are also subjects of fundamental importance for the "forma-mentis" of the student).

Response 7: The authors recognize that teaching a Fab-Lab methodology requires an investment in time, people, and money. Within the context of this study, the majority of the digitally fabricated tools were pre-manufactured for the teachers partially to deal with the time limit of conducting a study with working professionals during the school year. However, historically in the North American context, subjects such as woodshop were conventionally part of formal K-12 education and only recently eliminated due to a focus on standardized testing and a negative stigma associated with blue-collar jobs. Within the context of this study, we do not view these lessons to be consistent with the open-ended exploration that one might find while constructing in a Fab-Lab. Rather, this study is expanding upon the current notion of mathematics to move away from the pure number and symbol manipulation towards the exploration of connections between nature and the arts through the digitally fabricated tools. In other words, we want teachers and students to realize that there are other innovative ways to learn math and explore the wonder, joy, and beauty of mathematics.

Point 8: One of the main problems that Fab-Lab proposers have had to face in secondary schools in Chile is that, actually, few elementary and secondary teachers are qualified to teach in the areas ranging from math, nature, to art. The same problem arises in the present work. How do the authors intend to tackle and solve this problem? By organising a series of training courses for teachers?

Response 8: The authors agree that there is currently a lack of training to use digital fabrication at the primary and secondary levels. Our participants show very high motivation and interest to learn more about computer-aided design and digital fabrication and strongly suggested we offer follow-up workshops and make them into a series. Furthermore, they suggested that we can bring the workshops to schools and work with teachers and students. Yes, we are planning a follow-up study where we will be training teachers from the ground up on how to design patterns using software such as Adobe Illustrator, Inkscape, and/or Corel Draw. As a starting point, teachers will learn how to accurately replicate geometric patterns from Ernest Irving Freese’s Geometric Transformations. Additionally, a separate project (unrelated to the authors) is working to make a CAD repository of teacher resources and related lesson/project plans that will be hosted by the journal Contemporary Issues in Technology and Teacher Education (CITE) and/or a newly established journal, Journal of Technology Integrated Lessons and Teaching, where we hope to host exemplary resources and lesson plans created by the teachers.

Point 9: Apart from the technical problems, there are also problems of a purely administrative and legal nature to be solved. More specifically, Europe is actually facing the problem of "Unification and Harmonisation" of the study courses and diplomas. This should apply to the Countries of West Europe as well as the Countries of Eastern Europe. Now, if the Fab-Lab method were to be adopted at a European level to primary and secondary schools, there would be no "National standards" or "Teacher certification" in this case. What suggestions can the authors provide on this issue?

Response 9: The authors believe that movements towards standardization and unification are outside of the scope of this study. High-quality mathematic manipulatives can align with standards; thus, we do not believe standardization impedes the use of a digitally created tool. We openly acknowledge that “making” has traditionally thrived in more informal learning environments, and unfortunately, national standards may continue to exasperate that trend. The tension between Agency and Structure (what Keith Sawyer refers to as “the teaching dilemma”) has been around for a long time and the authors do not claim to have a simple answer for such a complex problem. However, with ample professional learning and targeted instructional coaching, teachers have the ability to design integrated making projects and align maker education with content area standards, which is an approach advocated by some maker education scholars in the U.S.

Point 10: Last but not least, the economic aspect must also be considered. Indeed, the Fab-Lab methodologies can be costly (ref., for instance to Forside - fablabFabLab Kambsdalur, etc.). Many countries (for example several countries of Latin America and possibly other countries) are not able to support these expenses at national level, but only for some schools and only on an experimental basis. This would lead to a divergence at the educational level between "rich" and "less rich" countries. What is the opinion of the authors on this?

Response 10: We agree that economic considerations are important when considering the implementation of these methodologies. In the case of this study, the researchers had access to a $5,000 laser-cutter which is not available at most primary and secondary schools. As such, the follow-up study plans on also incorporating lower-tech tools such as Silhouette Cameo cutters, which can be purchased in the $300-500 range, or a Cricut machine, which is in the $180-430 range. Additionally, as demonstrated in the post-survey responses from participants, multiple teachers took the tools back to their classroom and hand-cut paper sets of the tools so they could have a class set. The authors applaud the efforts of the Fab Lab Network for bringing these technologies to communities around the globe. The field of education has long recognized the Matthew Effect – where the rich get richer and the poor stay poor. However, the authors are happy to see that tools such as consumer-grade 3D printers have become much more affordable and are highly reliable. Ultimately, we hope to see students take on a mindset as a creator rather than a consumer, and we believe that the democratization of these technologies will continue to increase the accessibility to all teachers and students in the years and decades to come. As we mentioned before, in the follow-up study and other potential projects, we will continue to examine how schools integrate and implement these technology innovations.

Point 11: Finally, considering the answers to the above questions, can the authors insert a short paragraph specifying the added value of the present work compared to similar initiatives currently proposed and reported in the literature?

Response 11: Two paragraphs have been added as a new section, 5.3 Added Value of the Present Work. Please read the highlighted text in the manuscript and thank you for all your feedback.

Conclusions

Point 12: As said, the work is interesting and it was a pleasure to read it. In my opinion it deserves to be published. However, many works, initiatives, and workshops concerning the Fab Lab methodologies, based on digitally fabricated mathematics tools, have appeared in the literature (for example, in the EU there are several EC sub-programs entirely dedicated to this topic). In order to stimulate and attract the interest of the reader more, it is therefore advisable, in my opinion, to show the added value brought by this work compared to similar and existing initiatives, maybe with the help of the answers to the questions raised above.

Response 12: Thank you for the compliments. We are happy to include more citations to any of the works that you recommend, however, we were having difficulty locating several of the Fab Lab projects that you mentioned. Following your recommendation, we added two paragraphs as a new section, 5.3 Added Value of the Present Work. Please read the highlighted text in the manuscript and thank you for all your feedback.

Reviewer 3 Report

The authors pretend with this study examine how digitally fabricated mathematics tools can be used to help in-service teachers explore the wonder, joy, and beauty of mathematics through the authentic connections that exist between mathematics, nature, and the arts.

I suggest to the authors a clarification of the objectives of the mathematical content to be addressed and, mainly, an objectification of the results obtained compared to a control group where these tools were not used.

Author Response

Dear editor and reviewers:

Thank you so much for your feedback on our manuscript. We have revised the paper according to your suggestions and provided detailed explanations in the tables below. We appreciate your support and help to make our paper stronger.

Point 1: I suggest to the authors a clarification of the objectives of the mathematical content to be addressed and, mainly, an objectification of the results obtained compared to a control group where these tools were not used.

Response 1: Thank you for your comment. Our study was conducted from February to March 2022, when people were still worried about COVID-19. Therefore, it was the authors’ ethical consideration to only offer the workshops to the treatment group. We agree that it will be beneficial to have a control group in the future when the pandemic ends. Based on the finding from this study and the interest of our participants, we have planned a follow-up study that will more closely examine the integration question with the teachers learning how to use CAD to design, produce, and implement tools in their classrooms using a control-treatment design. Another project will be examining the impact of maker math tools on students’ achievements on special mathematical content and objectives with a control-treatment design. We have planned these follow-up studies and thank you for pointing out and confirming the future directions for us.

Round 2

Reviewer 2 Report

As mentioned in my previous report, the work is interesting and it was a pleasure to read it. Furthermore, the authors have satisfactorily answered all the questions raised in my previous review. The enthusiasm and passion shown by the authors in this matter are highly appreciated. In my opinion, this kind of work should be encouraged. In my opinion, this revised version deserves to be published.

Reviewer 3 Report

The authors, in their response, agree on the need for a control group for this project. In my opinion they cannot draw conclusions without this reference. Thus, I consider this work incomplect  and, unfortunately , I do not recommend it for publication.
